# Design and Measurement of a Two-Dimensional Beam-Steerable Metasurface for Ka-Band Communication Systems

David Rotshild [1],*, Daniel Rozban [1], Gil Kedar [2], Ariel Etinger [2] and Amir Abramovich [1]

1 Department of Electrical and Electronic Engineering, Ariel University, Ariel 40700, Israel; rozbandaniel@gmail.com (D.R.); amir007@ariel.ac.il (A.A.)
2 Ceragon Networks Ltd., Rosh Ha'ayin 4810002, Israel; gilke@ceragon.com (G.K.); ariele@ceragon.com (A.E.)
* Correspondence: drotshild@gmail.com

**Abstract:** This study introduces a steerable metasurface reflector designed for the Ka-band, enabling one-dimensional and two-dimensional beam steering. The paper elaborates on the design considerations, manufacturing process, and experimental findings. The unit cell design incorporates a Varactor diode as the tuning element, facilitating a dynamic phase range exceeding 300° with minimal metasurface beam steering losses. Notably, the experimental results are in good agreement with the simulation outcomes. The advantages of employing this metasurface reflector include rapid beam steering, cost-effective production implementation, support for both one-dimensional and two-dimensional beam steering, low reflection loss, high-resolution beam steering, and continuous beam steering capabilities.

**Keywords:** beam steering; Ka-band; metasurface; tunable circuits; devices

## 1. Introduction

Deploying wireless communication networks and access points for 5G/6G will require antennas with accurate and real-time steering capabilities due to the increase in transmitted frequencies towards the MMW band (30 GHz–300 GHz) [1]. Furthermore, MMW frequencies have lower diffraction characteristics, decreasing quickly with increasing distance, and require highly directional antennas to focus and steer their power [1,2]. Blockages or obstacles can cause differences in the MMW path loss characteristics. These paths are classified as the line-of-sight (LOS) and non-LOS, which need real-time steering and antenna corrections to optimize the channel [3,4]. One up-and-coming technology to realize such accurate steering and focusing capabilities and reconfigurability utilizes a unique MMW antenna based on a surface composed of an array of unit cells known as a metasurface (MS) [5].

MS, including reconfigurable MS, has evolved in recent years due to its unique electromagnetic properties, which are not found in nature [5,6]. Many studies and papers focusing on different MS applications covering various frequency bands and methods have been published. For example, applications in which MS plays an important role include cloaks [7], flat polarization control [8], and focusing and collimation flat reflect arrays [9]. Furthermore, interdisciplinary reconfigurable MS applications were demonstrated, such as reconfigurable MS antenna [10–12] and the extraordinary sensitivity enhancement detection of antibiotics [13].

The tunability and reconfigurability of MS can be realized using electronic components or materials such as PIN switches, MEMS switches, liquid crystals, and varactor diodes [14–20]. PIN switches enable a fast-switching time, high reliability, low cost, and low power consumption [14,15]. MEMS switches attracted attention recently due to their small size and linearity [16], but they have a relatively slow switching time compared to PIN switches and a limited lifetime. However, the disadvantage of switching elements

is that they prevent the continuous tuning of the MS unit cell's electromagnetic properties, such as their phase or resonance frequency. The Liquid Crystals (LC) method can be realized in MS as a tuning dielectric substrate without any external element, but it is more challenging to implement [17,18]. Furthermore, the LC has a slower response time than the PIN switch and varactor diode. Hence, LC is unsuitable for continuous tracking and switching applications, such as those required for communication channels needing ongoing optimization, particularly with the increasing frequency of the MMW band [18]. Those disadvantages also make it difficult to perform accurate two-dimensional (2D) beam steering with a high gain, low loss, and sidelobe-level suppression. On the other hand, the advantages of the varactor diode are continuous and include its high capacitance dynamic range, constant gamma for linear tuning, low loss, convenient PCB assembly, and meager power consumption [10–12,19–21]. This work discusses and analyzes the tuning element losses, substrate losses, DC control, and the isolation between the RF signal and DC. Increasing the isolation between RF and DC signals is important for better reflector performance. This was realized using a radial stub with a careful design.

Furthermore, a significant reduction in about 75% of analog channels was obtained using the Unit Cell Binning (UCB) method, in which we excite a group of Varactors with the same DC voltage. Using UCB, 2D beam steering was measured, achieving excellent agreement with the results obtained without using UCB. Another technology with a high advantage in RF isolation is Steer–by–Image–Technology (SBI), which uses an image to steer a beam rather than a DC circuit control, analog inputs, and a microcontroller/FPGA board [22–24].

The design considerations, manufacturing, and experimental testing of reconfigurable Ka-band MS are presented in this study. The use of two-dimensional beam steering for a variety of angles and several frequencies is demonstrated. This research is based on our previous study, in which we conducted experimental studies in which we demonstrated one-dimensional (1D) beam steering at 21–28.5 GHz [19].

## 2. Unit Cell and MS Reflector Design

The unit cell design and its equivalent RLC circuit model are shown in Figure 1.

The unit cell's front-side, back-side, profile, and 3D inside view are shown in Figure 1a–d, respectively. The unit cell comprised two commercial dielectric substrates of Rogers company (Chandler, AZ, USA) model RT/Duroid 4350B with a low dielectric constant of $\varepsilon_r = 3.66$ [25]. Prepreg RO4450F (Rogers company, Chandler, AZ, USA) was used to glue them together. Two vertical conductor strips were in the top layer, and the varactor diode was placed between them, as shown in Figure 2a. Varactor diode model MAVR-011020-1411 of MACOM (Lowell, MA, USA) was used in this unit cell design since it provides an extremely low capacitance, enabling a wide dynamic capacitance range. The varactor diode capacitance $C_d$ is between $C_{max} = 0.22$ pF and $C_{min} = 0.035$ pF for 0–15 V reverse bias voltages, respectively, providing a capacitance ratio of 7 [26].

The DC bias $V_d$ was connected to the anode of the varactor diode by a via, as shown in Figure 1d. A radial stub was used as the RF choke to isolate the DC bias from the RF. The middle copper layer was used as an RF ground, enabling DC control on the $C_d$, using a via and clearance.

Figure 1e shows the unit cell equivalent circuit model, which includes the $C_d$ and the intrinsic electrical parameters of the unit cell. $C_{int}$ is the parasitic capacitance, and $L_{int}$ is the unit cell's parasitic inductance due to its structure and geometry. $R_{diel}$ and $R_{ohm}$ are the dielectric and ohmic resistances of the unit cell, respectively [19,27]. The MS reflector structure with the DC bias scheme is shown in Figure 2.

Figure 2 shows the MS structure, including the simple DC bias circuit, which reduces the complexity and absorption of the unit cell. The continuous strip on the top layer of unit cells is used as the DC ground per each MS column. The dashed strip is connected to a separate DC bias using the via for each unit cell. Thus, the DC bias controls the phase shift of each unit cell in the MS independently, enabling 2D steering.

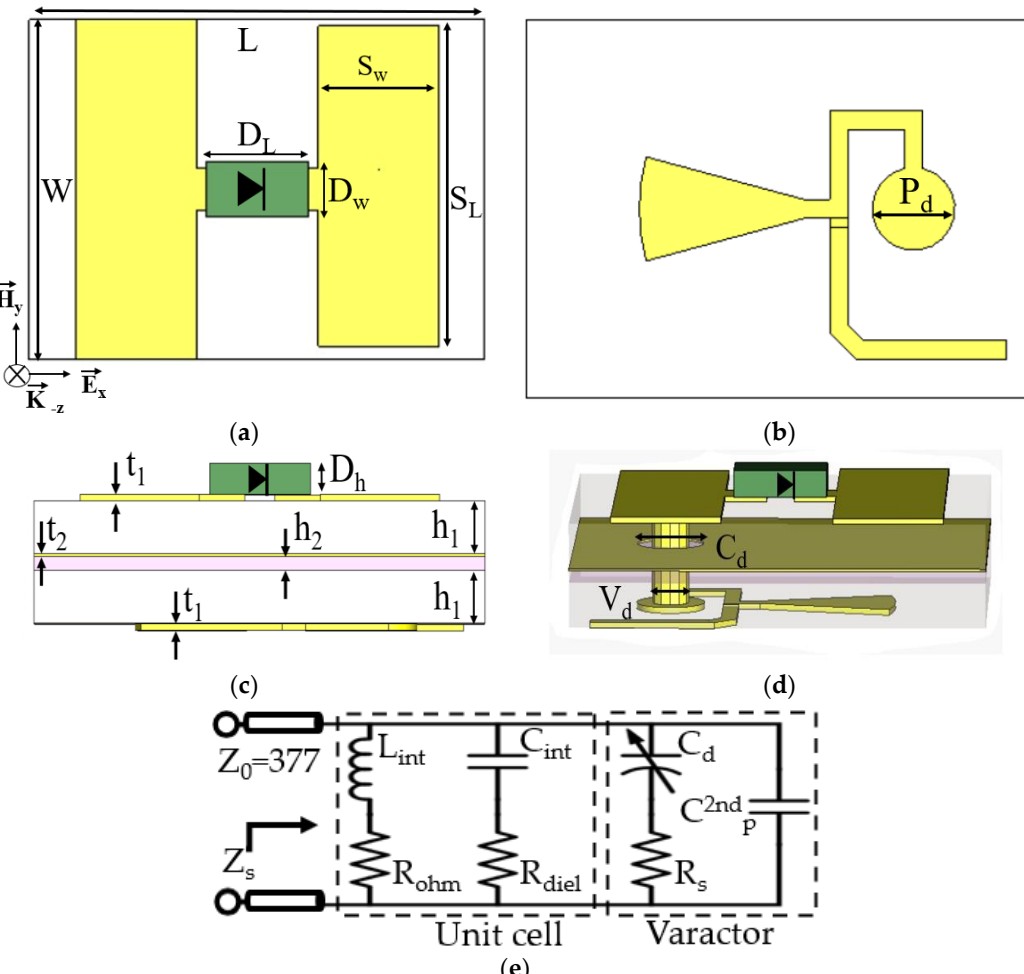

**Figure 1.** Unit cell design: (**a**) unit cell frontside; (**b**) unit cell backside; (**c**) profile view of the unit cell; (**d**) 3D inside view of the unit cell; (**e**) unit cell's equivalent RLC circuit model.

**Figure 2.** MS structure with varactors and DC bias idea: (**a**) MS frontside; (**b**) MS backside with varactors DC bias indexes; (**c**) 3D inside view of MS closure of the electrical circuit for each varactor.

The MS was designed to work at 21–28.5 GHz for incident polarization, as shown in Figures 1a and 2a. The final optimized unit cell parameters, which follow the standard PCB manufacturing process, are summarized in Table 1.

**Table 1.** Unit cell parameters.

| Parameter | Description | Value [mm] |
|---|---|---|
| $L$ | Unit cell length | 3.4 |
| $W$ | Unit cell width | 2.55 |
| $S_L$ | Pad/line length | 1.4 |
| $S_w$ | Pad/line width | 0.9 |
| $S_g$ | Vertical gap between the slots | 0.15 |
| $D_L$ | Varactor diode length | 0.7615 |
| $D_w$ | Varactor diode width | 0.406 |
| $D_H$ | Varactor diode height | 0.203 |
| $h_1$ | Dielectric substrate thickness | 0.338 |
| $h_2$ | Prepreg substrate thickness | 0.102 |
| $t_1$ | Copper thickness | 0.043 |
| $t_2$ | Copper thickness | 0.018 |
| $P_D$ | Via pad diameter | 0.55 |
| $V_D$ | Via diameter | 0.3 |
| $C_D$ | Clearness diameter | 0.55 |

## 3. Simulations and Experimental Results

The TEM floquet port of CST 2023 simulation software [28] was used to analyze the proposed unit cell's reflection. Figure 3 shows the CST simulation results compared to the reflection measurements of the designed unit cell. The comparison was measured for different capacitance values of, $C_d$, determined by the reverse bias voltage, $V_d$. The experimental setup is shown in Figure 3c.

The total resistance $R$ of the designed unit cell influences the reflected beam magnitude intensity and the phase. The total resistance, $R$, is composed of internal and external resistances, as shown in the unit cells' equivalent RLC circuit model presented in Figure 1e [19]. The dielectric losses $R_{diel}$ and the ohmic losses $R_{ohm}$ are intrinsic resistances, where $R_s$ is the varactor serial resistance. The values of $R_{diel}$ and $R_{ohm}$ can be well-quantified by the CST simulation, while the value of $R_s$ depends on the components' assembling quality. Based on similar studies in the literature, it was found that the value of $R_s$ is 5 Ω [29,30], with good agreement between the simulation and the measurements.

**Figure 3.** *Cont.*

**Figure 3.** Simulation and measurement results of unit cell reflection for several $C_d$ values: (**a**) magnitude; (**b**) phase. The dashed black lines are the simulation results, and the solid red lines are the experimental results. (**c**) The reflection experimental setup.

Two wide-band, double-ridge horn antennas, model PowerLOG 40400, from Aaronia Company (Strickscheid, Germany) [31], were used as RX and Tx for the RF signal. The two antennas were placed perpendicular to the MS. One was used to transmit the radiation, and the second was used to receive the reflected signal from the MS. Both were connected to a Vector Network Analyzer (VNA) of Keysight Company (Santa Rosa, CA, USA) model E8364A. The experimental reflection measurements of the MS were normalized to the reflection measurements of a copper plane surface with the same dimensions. The dynamic reflection phase range of the proposed unit cell at 28 GHz as a function of $C_d$ for simulation and $V_d$ for measurement is shown in Figure 4.

**Figure 4.** The reflected phase dynamic range of the unit cell at a frequency of 28 GHz is a function of the varactor capacitance $C_d$ in the black line (simulation results) and a function of the varactor reverse voltage $V_d$ in the red line (experimental results).

The phase dynamic range values shown in Figure 4 were normalized between 0° and 360° with a value of 0° for $C_{d\ max}$ ($V_d$ = 0 V) and 301° for $C_{d\ min}$ ($V_d$ = 16 V). These results show excellent agreement between the simulation and measurement of the phase dynamic range. Note that the simulation result is for an infinite surface while the experimental result is extracted from the final size of the MS prototype with the influence of the edges' effect.

### 4. Theory of 2D Steering of MS Reflector

According to the phased array theory, each unit cell location on the MS is defined at its center [32]. A 2D surface on the *XY* plane with the spatial array arrangement of a fixed distance and a 90° angle between the unit cells is defined as $S(x_j, y_i)$, $j = 1, 2,\ldots, M$ and $i = 1, 2,\ldots, N$. When both $N$ and $M$ are integers, this results in an array containing $N \times M$ elements. The MS reflector side's cross-section view scheme is presented in Figure 5.

**Figure 5.** Steering properties of 2D reconfigurable reflective surface for incident rays $L_0$-$L_N$, normal to the MS.

Figure 5 shows a reconfigurable MS reflector scheme. $L_1$, $L_2$, and $L_N$ are incident rays towards the surface. Due to a planned gradual phase provided by the unit cells' MS reflector, the rays are reflected at the desired angle $\theta_x$. The Optical Path Difference (OPD) between the cells is defined as $\Delta L$ [9] and is shown in Equation (1)

$$\Delta L = \Delta X \cdot sin(\theta_x) \tag{1}$$

where $\Delta X$ is the array constant. The OPD in phase difference is shown in Equation (2):

$$\Delta\varphi_x = 360 \cdot \Delta L / \lambda \tag{2}$$

A gradual accumulating phase, obtained by supplying a phase difference $\Delta\varphi_x$ between adjacent unit cells $x_{(j)}$ and $x_{(j+1)}$ in $\hat{x}$ axis, yields the desired steering angle $\theta_x$ in the *XZ* plane. Using Equations (1) and (2), the connection between $\Delta\varphi_x$ and $\Delta x$ and the desired angle steering $\theta_x$ is shown in Equation (3):

$$\theta_x = sin^{-1}(\lambda \cdot \Delta\varphi_x / 360 \cdot \Delta X) \tag{3}$$

For the steering angle $\theta_y$ in the YZ plane, the same analysis applies to $y_{(i)}$ along the $\hat{y}$ axis when utilizing $\Delta\varphi_y$ and $\Delta y$ in Equations (1)–(3). A superposition of independent steering in planes *XZ* and *YZ* results in 2D beam steering ability $\theta$, as shown in Equation (4).

$$\theta = \sqrt{\theta_x{}^2 + \theta_y{}^2} \tag{4}$$

The simulation results demonstrating the realization of 2D beam steering using the simultaneous combination of two orthogonal beam steering in 1D are shown in Figure 6.

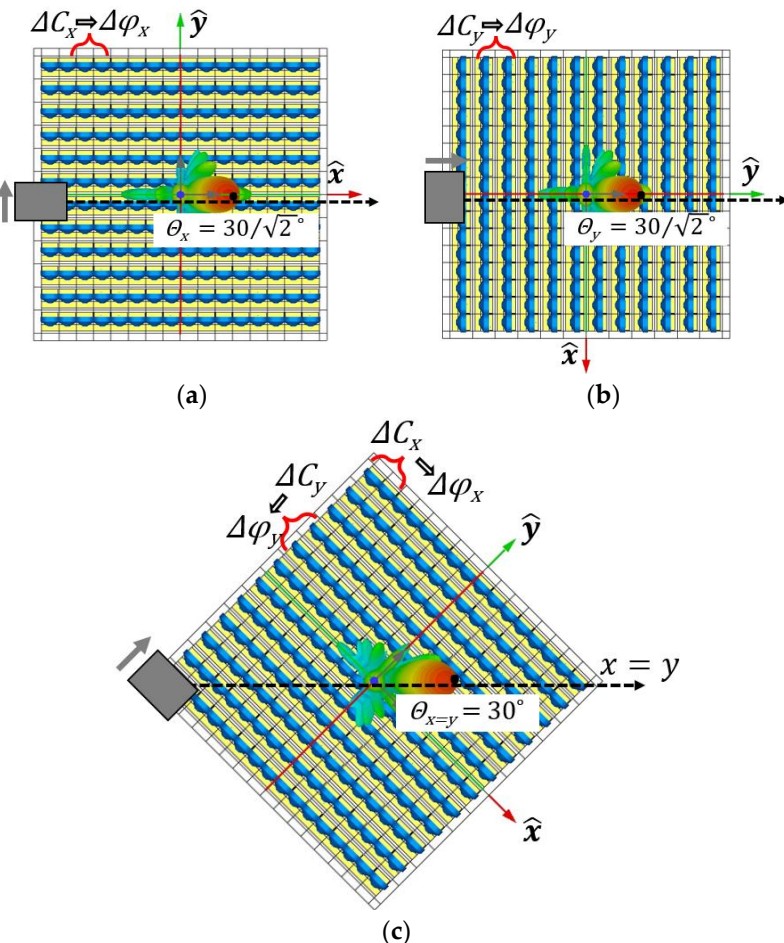

**Figure 6.** Combining two orthogonal 1D beam steerings in $\theta_x$ and $\theta_y$ into 2D beam steering: (**a**) 1D beam steering $\theta_x$; (**b**) 1D beam steering $\theta_y$; (**c**) 2D beam steering $\theta_{x=y}$. The gray rectangle and arrow represent the antenna and the polarity of the incident electric field, respectively.

Equation (3), which connects the designed $\theta$ to the $\varphi$ and the black curve in Figure 4, determines each varactor's capacitance $C_d$.

For $\theta_x = 30°/\sqrt{2}$, $\varphi_x = 41°$ is applied cumulatively along the columns, leading to the result shown in Figure 6a. For $\theta_y = 30°/\sqrt{2}$ the $\varphi_y = 31°$ is applied cumulatively along the rows leading to the result shown in Figure 6b.

It is necessary to pay attention to the MS orientation, which changes between Figure 6a,b by 90° degrees, and between Figure 6c by 45°. The combined beam steering on the $\hat{x}$ axis and the $\hat{y}$ axis simultaneously allows for the 2D beam steering, as shown in Figure 6c. This leads to the beam steering angle at $\theta_{x=y} = 30°$, following Equation (4).

The 2D phase distribution throughout 12 columns and 16 rows of the MS, which leads to the simulation result shown in Figure 6c, is presented in Table 2.

Table 2 shows the ideal values in the wrap phase mode range of 0–360°. As shown in the table for the consecutive columns, the condition of $\varphi_y = 41°$ is met, and for the consecutive rows, the condition of $\varphi_x = 31°$ is met, leading to $\theta_y = \theta_y = 30°/\sqrt{2}$. The results in Figure 6 are in full compliance with the calculations for both the 1D and the 2D results. A value that exceeds 301°, which is the maximum of the dynamic reflection phase range, is rounded to the nearest existing value; for example, the value in column 9 and row 1 is set to 301° and the value in column 9 and row 2 is set to 0°. A high dynamic reflection phase range allows the MS spatial phase distribution to be considerably changed, where the dynamic phase is used cyclically as needed. These MS properties are frequency-sensitive and allow for the steering of the incident beam in a limited frequency band around the

working frequency. Steering at other supported frequencies will require using different dynamic phase ranges.

**Table 2.** Phase distribution throughout 12 columns and 16 (°).

|  | 1 | 2 | 3 | 4 | 5 | 6 | 7 | 8 | 9 | 10 | 11 | 12 |
|---|---|---|---|---|---|---|---|---|---|---|---|---|
| **1** | 0 | 41 | 82 | 123 | 164 | 205 | 246 | 287 | 328 | 9 | 50 | 91 |
| **2** | 31 | 72 | 113 | 154 | 195 | 236 | 277 | 318 | 359 | 40 | 81 | 122 |
| **3** | 62 | 103 | 144 | 185 | 226 | 267 | 308 | 349 | 30 | 71 | 112 | 153 |
| **4** | 93 | 134 | 175 | 216 | 257 | 298 | 339 | 20 | 61 | 102 | 143 | 184 |
| **5** | 124 | 165 | 206 | 247 | 288 | 329 | 10 | 51 | 92 | 133 | 174 | 215 |
| **6** | 155 | 196 | 237 | 278 | 319 | 0 | 41 | 82 | 123 | 164 | 205 | 246 |
| **7** | 186 | 227 | 268 | 309 | 350 | 31 | 72 | 113 | 154 | 195 | 236 | 277 |
| **8** | 217 | 258 | 299 | 340 | 21 | 62 | 103 | 144 | 185 | 226 | 267 | 308 |
| **9** | 248 | 289 | 330 | 11 | 52 | 93 | 134 | 175 | 216 | 257 | 298 | 339 |
| **10** | 279 | 320 | 1 | 42 | 83 | 124 | 165 | 206 | 247 | 288 | 329 | 10 |
| **11** | 301 | 351 | 32 | 73 | 114 | 155 | 196 | 237 | 278 | 319 | 0 | 41 |
| **12** | 341 | 22 | 63 | 104 | 145 | 186 | 227 | 268 | 309 | 350 | 31 | 72 |
| **13** | 12 | 53 | 94 | 135 | 176 | 217 | 258 | 299 | 340 | 21 | 62 | 103 |
| **14** | 43 | 84 | 125 | 166 | 207 | 248 | 289 | 330 | 11 | 52 | 93 | 134 |
| **15** | 74 | 115 | 156 | 197 | 238 | 279 | 320 | 1 | 42 | 83 | 124 | 165 |
| **16** | 105 | 146 | 187 | 228 | 269 | 310 | 351 | 32 | 73 | 114 | 155 | 196 |

The marked values are values that exceed 301°.

To ensure the feasibility and simplicity of the far-field measurement setup in this work, we defined a fixed measurement cross-section, shown with a dashed black arrow in Figure 6a–c, according to the constraints of the stepper motor and the optomechanical equipment so that, for a change in the direction measurement from $\theta_x$ to $\theta_y$ or $\theta_{x=y}$, where $X = Y$, it is only necessary to rotate the MS and antennas (Tx and Rx), which are shown as a gray rectangle with an arrow representing the polarization.

## 5. Beam Steering Experimental Results

The final design of the MS reflector prototype was based on the unit cell and MS configurations described above (see Figures 1 and 2). The manufactured prototype, including its peripheries, is shown in Figure 7.

**Figure 7.** Photo of the manufactured MS prototype: (**a**) front side; (**b**) backside; (**c**) GUI panel with an example; (**d**) microcontroller and DAC channels; (**e**) cables with 192 DC channels connected to the back side of the MS reflector.

The MS prototype size is 40.8 mm × 40.8 mm with 16 rows and 12 columns of unit cells, totaling 192 unit cells. Six DACs of Analog devices (Wilmington, MA, USA) model

AD5373 were used to drive all the varactors of the MS. A Raspberry Pi (Cambridge, UK) zero development board was used to control the DAC modules to generate the required voltage for each varactor of the MS unit cells. A software algorithm was used to calculate the required voltage as the function of the required phase. Figure 4 shows the reflected phase curve in red as the function of the reverse varactor voltage, $V_d$.

The far-field measurements' concept and setup are shown in Figure 8.

**Figure 8.** Far-field measurement setup: (**a**) scheme for sample acquisition $\theta_r = 0°$; (**b**) scheme for sample acquisition $\theta_r = 30°$; (**c**) A photo of the far-field K-band experimental setup. The yellow dashed arrows show the path of the Tx antenna, which moves along with the MS. The solid yellow arrows indicate the direction of the stepper motor's motion. Consequently, the MS is positioned at the center of the Tx antenna's circular trajectory.

A programable stepper motor was used to rotate the Tx antenna while the Rx antenna was fixed (see Figure 8). The values of $\theta$ are related to the MS normal, defined as $\theta_r = 0°$ in the far-field experimental setup shown in Figure 8a. For example in Figure 8a the normal incident case, and in Figure 8b the acquisition at $\theta_r = 30°$, both the MS and the Tx rotate to negative $\theta = -30°$, with the angular axis marked by the black dashed line, where the receiver is fixed in position. The Tx is placed on the ruler in front of the MS, and the step

motor is used to rotate the MS and Tx simultaneously, as shown in Figure 8. Figure 8c is a photo of the far-field K-band experimental setup where the MS and the horn antennas were rotated by 45°; this setup is used for 2D far-field steering measurements, as illustrated in Figure 6.

Simulation results for the whole far-field pattern for the uniform reflected phase of the MS structure, and reference results for the calibration and validation of the experimental setup, are shown in Figure 9.

Figure 9a shows the full far-field pattern of the MS-reflected beam to better understand the far-field pattern and the MS orientation relative to the defined steering coordinates $\theta_y$ and $\theta_x$, shown as black arrows, where the blue components represent the varactor diodes. To steer the beam on the $\theta_x$ coordinate, namely the Azimuth axis, it is necessary to gradually change the reflected phase in the x-axis of each column by $\Delta\varphi_x$. To steer the beam in the $\theta_y$ coordinate, namely the elevation axis, one needs to gradually change the reflected phase of each row by $\Delta\varphi_y$.

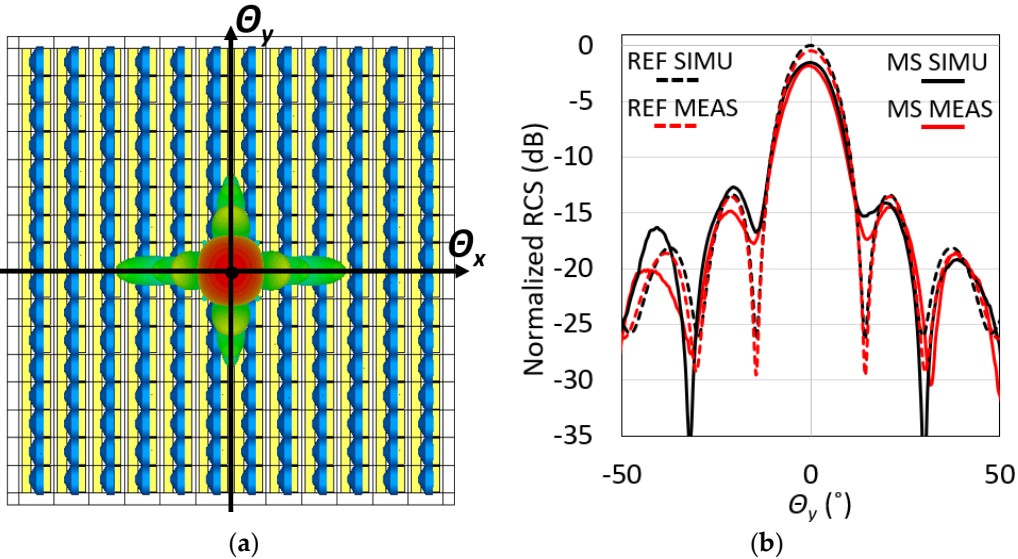

(**a**)  (**b**)

**Figure 9.** Full 3D far-field simulation and reference results: (**a**) full far-field simulation results relative to the designed MS with defined beam steering coordinates; (**b**) comparison between simulations and measurement results for the reference cases at cut plan $\theta_y$, where $\theta_r = 0°$.

Figure 10b shows the far-field simulation and measurement results of the MS for the case of a uniform reflected phase for all unit cells ($\theta_r = 0°$) compared to the reference copper plate of the same size as the MS.

The main lobe and the second side lobe's simulation results show good agreement with the experimental results. As expected, the loss of the MS is slightly higher than that of the copper surface. These results validate the use of the configuration of the experimental setup to measure the far-field pattern of the MS-reflected beam.

Using the experimental setup shown in Figure 8c, the beam steering performance of the designed MS compared to the CST simulation results was obtained. The polarization of the incident wave is described in Figure 5. The simulation and experimental results of 1D steering for several $\theta_x$ angles and $\theta_y$ angles for 28 GHz are shown in Figure 10.

The results in Figure 10 show good agreement with the measurements and the simulation results, including the decreasing trend in the reflected beam intensity as $\theta_r$ increases. The main loss factor of this reconfigurable MS reflector which is the varactor diode, is to be further explained later on.

Simulation results are compared to the experimental results of 2D beam steering for several $\theta_{x=y}$ angles at 28 GHz are shown in Figure 11.

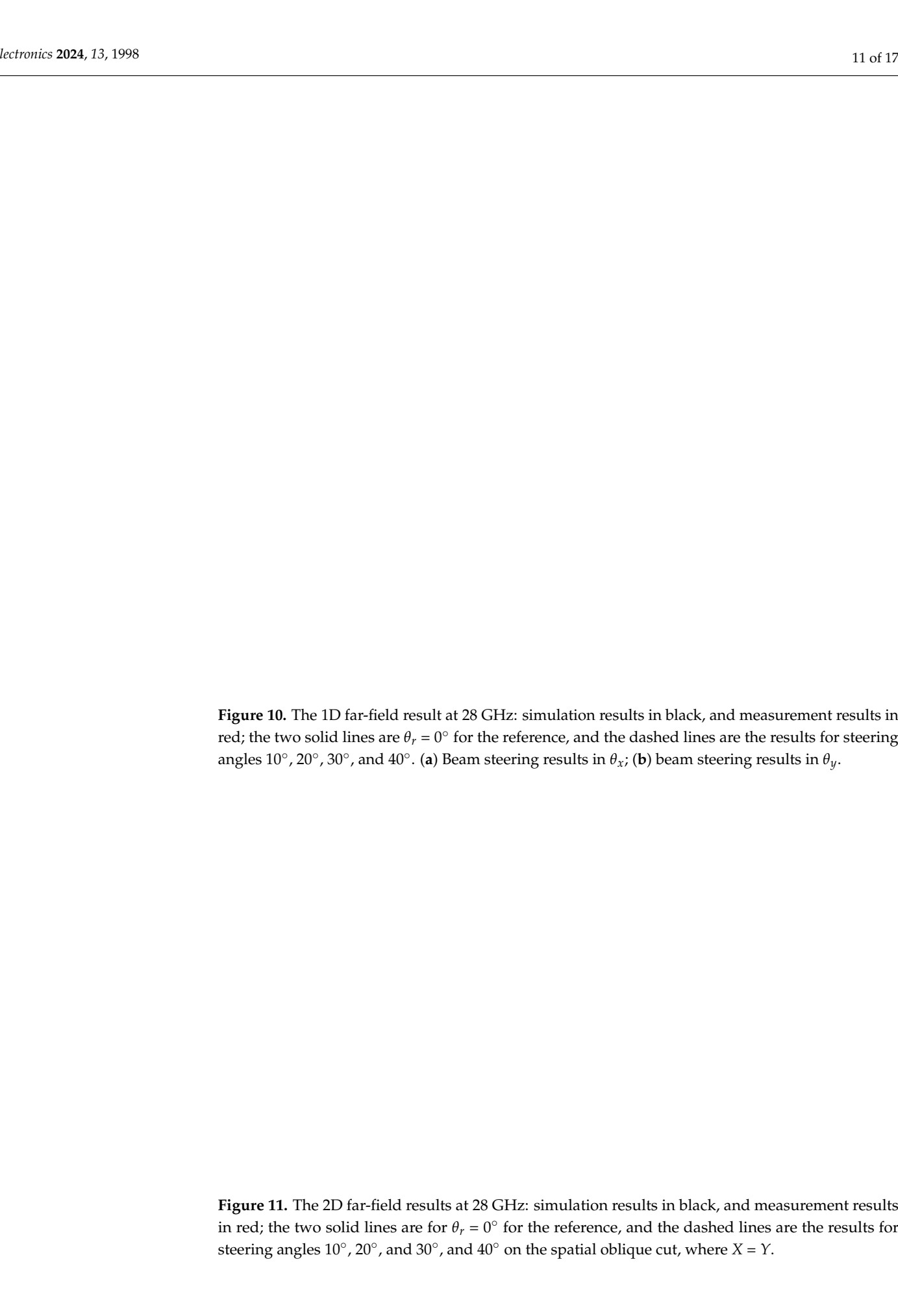

**Figure 10.** The 1D far-field result at 28 GHz: simulation results in black, and measurement results in red; the two solid lines are $\theta_r = 0°$ for the reference, and the dashed lines are the results for steering angles $10°$, $20°$, $30°$, and $40°$. (**a**) Beam steering results in $\theta_x$; (**b**) beam steering results in $\theta_y$.

**Figure 11.** The 2D far-field results at 28 GHz: simulation results in black, and measurement results in red; the two solid lines are for $\theta_r = 0°$ for the reference, and the dashed lines are the results for steering angles $10°$, $20°$, and $30°$, and $40°$ on the spatial oblique cut, where $X = Y$.

The 2D beam steering simulation results were compared to experimental results for about 10°, 20°, 30°, and 40° on the spatial oblique cut, where $X = Y$, designed by simultaneously combining equal beam steering in $\theta_x$ and $\theta_y$ according to Equation (4). These were then compared to reference results at $\theta_r = 0°$ at 28 GHz, as shown in Figure 11. A good agreement between the measurements and the simulation results is shown with the pattern of the results, $\theta$ position of each beam steering result, and the power decrease trend as $\theta$ increases. The technique used to measure the 2D beam steering is explained above and shown in Figure 6, and the far-field experimental setup is shown in Figure 8c.

The comparison between the simulation results and the experimental data presented in Figures 10–12 reveals disparities in the gain and steering angle, and certain distortions within the measured results.

**Figure 12.** The 2D far-field beam steering measurements at 27 GHz, with steering angles approximately at 10°, 20°, 30°, 40° and 45° along the spatial oblique cut where $X = Y$, are represented by the dashed curves. The solid line is a reference for $\theta_{x=y} = 0$.

The primary factor contributing to these variances stems from the varactor circuit model, outlined by the varactor equivalent circuit depicted in Figure 1e. The series resistance $Rs$ of the varactor primarily contributes to the decrease in the magnitude of the reflected beam. Furthermore, as the resonance frequency approaches the unit cell resonance, a higher loss in the unit cell reflection magnitude occurs, as shown in Figure 3a. The phase shift of each unit cell is determined according to the reflected phase curve depicted in Figure 4. This simulation was obtained by simulating the correlation between the varactor capacitance and the resulting reflected phase. Conversely, the experimental reflected phase curve was obtained by measuring the relationship between the reverse varactor voltage and the reflected phase. This variation could account for the discrepancy in the steering angle observed in Figure 10. Moreover, slight variations in the tolerances of the Varactor capacitance can lead to minor adjustments in the required phase shift. These subtle alterations in the phase shift can elucidate the differences observed in the beam steering angle between simulations and measurements. The phenomena mentioned above cause an attenuation as a function of the beam steering angle $\theta$ that does not decay monotonically proportional to $\cos(\theta)$ as was expected [32]. Similar decaying attenuation patterns have been observed in related studies [33–35]. It's worth noting that the primary factors influencing these mechanisms and behaviors are the lossy nature of the varactor and the resonance characteristics of the unit cell.

This MS supports beam steering at frequencies between 21 GHz and 30 GHz. The 2D beam steering ability is highlighted in this study, and thus, additional 2D beam steering results were demonstrated at 27 GHz, as shown in Figure 12.

The distortion observed in Figure 12 at a small angle can be attributed to the stronger coupling between the Rx and Tx antennas, especially in 2D steering where the antennas are closer to each other along a larger arc because they are diagonal, as shown in Figure 8c.

The proposed prototype can mitigate beam steering deviations by optimizing the varactor reverse voltage, which can be continuously adjusted with a resolution in the order of 0.4°, as was shown in our previous study [19]. An angle steering resolution in the order of 1° for the proposed prototype is illustrated in Figure 13b below.

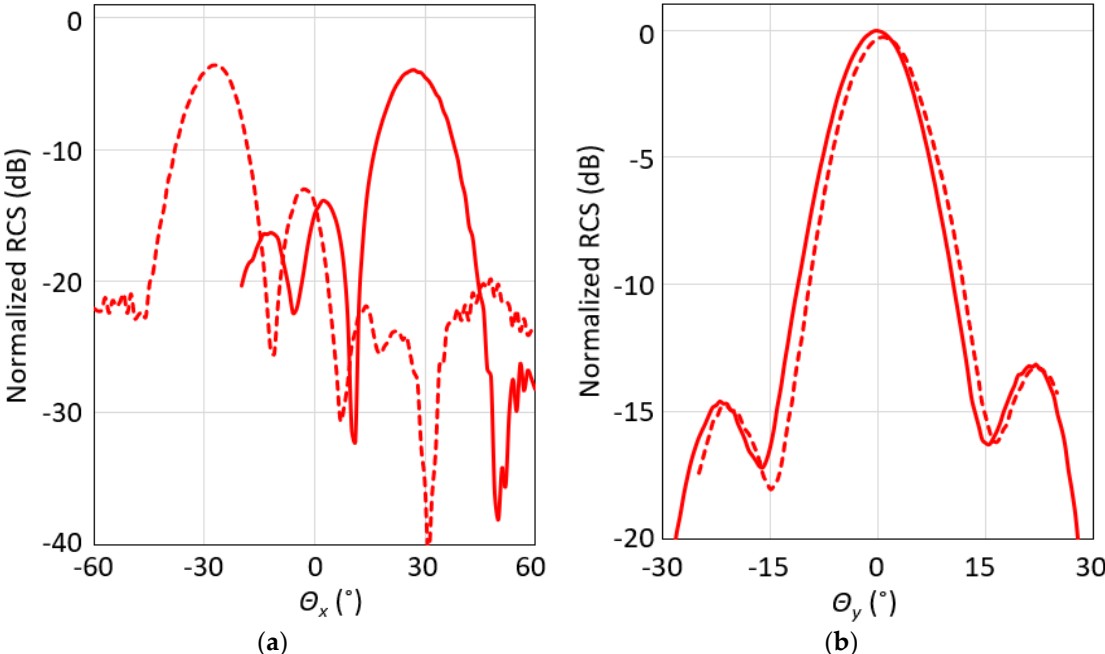

**(a)**                                                                 **(b)**

**Figure 13.** The 1D far-field beam steering results at 28 GHz. (**a**) The negative angle beam steering result represented by the red solid line is compared to the same $\theta$ angle's positive beam steering result, represented by the red dashed line, at the $\theta_x$ coordinate. (**b**) The beam steering result at $\theta = 1°$, represented by the red dashed line, is compared to the reference result $\theta = 0°$, represented by the solid red line, at the $\theta_y$ coordinate.

Additional experimental results demonstrate the angle symmetry properties and high beam steering angle resolution, which is shown in Figure 13.

The MS was designed using a quasi-symmetrical unit cell, enabling positive and negative beam steering angles. For simplicity, the results focus on demonstrating beam steering within the positive half of the $\theta$ angle axis, as illustrated in Figures 10–13. To validate symmetry, we adjusted the setup to measure beam steering at negative $\theta$ angles, confirming the similarity between positive and negative beam steering, as shown in Figure 13a. The ability to continuously control the phase at the unit cell level is demonstrated in Figure 13b, showcasing a high-beam steering angle resolution of 1°.

A significant reduction in the required analog inputs can be achieved if we connect the same DC voltage to a group of unit cells, referred to as Unit Cell Binning (UCB). Using a UCB of four, a 75% reduction in the number of analog inputs was achieved, as shown in Figure 14.

Figure 14 compares the typical case, where each unit cell is connected to a separate DC voltage (dashed line), to the four UBC cases (solid line) at 28 GHz. The experimental results were in excellent agreement in both cases and showed the high feasibility of using UBC algorithms for MS.

**Figure 14.** 2D far-field beam steering results at 28 GHz for several beam steering in $\theta_x$ axis and at a constant elevation ($\theta_y$.= 20°) with UBC, represented by a dashed red line, compared to that without UBC, represented by the red solid line.

The comparison between the use of several tuning methods for a tunable MS reflector is presented in Table 3.

**Table 3.** Comparison between MS reflectors.

| Reference | Frequency (GHz) | Steering Dimension | Phase Range (Deg) | Steering Range (Deg) | Tuning Technology | Switching Time | DC Bias Method |
|---|---|---|---|---|---|---|---|
| [33] | 28 | 2D * | 0 or 180 | 40 | PIN switch | Fast | Digital |
| [34] | 31 | 1D | 180 | 65 | Varactor | Fast | −20 to +10 |
| [35] | 37.5 | 2D | 338 | 35 | Liquid Crystal | Slow | 0 to +36 |
| This work | 28 | 2D | 301 | 45 | Varactor | Fast | 0–16 [V] |

* Beam steering in 2D was demonstrated but only separately [33].

Table 3 provides a performance comparison within the Ka-band. From this table, it is evident that this offers several advantages. These advantages include a theoretically continuous adjustment of the beam steering; however, only a few works in this frequency band, especially in the context of 2D beam steering, were demonstrated.

This work demonstrates the ability to change a small angle in the beam steering direction and reduce the number of control points on the unit cell phase with the help of binning. This adds to the simplicity of the production, which includes a standard PCB with a small number of layers that can be parametrically adjusted to a wide frequency range.

## 6. Conclusions

2D beam steering (Azimuth and Elevation) using an MS reflector Ku band was demonstrated experimentally. The experimental results were in good agreement with the simulation results. A high phase dynamic range of above 300° in the Ku-band, and a bandwidth of more than 4 GHz with relatively low losses, were demonstrated. 2D beam steering with an angle dynamic range of about ±45° was measured, showing excellent agreement with the simulation results. The proposed MS reflector achieves high resolution and high accuracy in terms of beam steering angle at a 1° angle resolution, as demonstrated in this work.

Furthermore, the UBC method significantly reduces the required DAC channels by 75%. Therefore, such MS sub-reflectors can be used in the Cassegrainian antenna configuration, yielding an improved performance compared to conventional Cassegrain antennas. The advantages of using an MS sub-reflector in a Cassegrain antenna are that it is lightweight, has an electrical remote control, has a low profile, has a fast response time, is simple to manufacture, is highly reliable, and is cheaper than mechanical steering.

A drawing using the MS reflector technology concept as a sub-reflector in a Cassegrain antenna is shown in Figure 15.

**Figure 15.** Basic Cassegrain antenna diagram with flat MS as a sub-reflector: (**a**) without beam steering ($\theta = 0°$); (**b**) with beam steering ($\theta = 10°$). The dashed red arrows indicate the radiation path from the antenna feed to the sub-reflector and then to the parabolic reflector. Beam steering from the parabolic reflector can be achieved by steering the beam at the sub-reflector.

In this configuration, the MS steers the beam from the feed and changes its direction, accordingly, improving the directivity, which becomes an issue in 5G point-to-point MMW communication [36]. Furthermore, the suggested configuration enables real-time control over the beam direction and increased stability, which are needed due to environmental influences, such as wind and temperature differences [34]. In this configuration, the MS steers the beam camming from the feed changing the beam direction, accordingly, improving the directivity, which becomes an issue in 5G point-to-point MMW communication [37]. Furthermore, the suggested configuration enables real-time control over the beam direction and increased stability, which are needed due to environmental influences, such as wind and temperature differences [34].

**Author Contributions:** Conceptualization, D.R. (David Rotshild); methodology, D.R. (David Rotshild), D.R. (Daniel Rozban), A.A. and G.K.; software, D.R. (Daniel Rozban); validation, D.R. (Daniel Rozban) and A.E.; data curation, D.R. (David Rotshild); formal analysis, D.R. (David Rotshild) and D.R. (Daniel Rozban); investigation, D.R. (David Rotshild), D.R. (Daniel Rozban) and A.A.; resources, D.R. (David Rotshild) and A.A.; writing—original draft preparation, D.R. (David Rotshild) and A.A.; writing—review and editing, D.R. (Daniel Rozban), A.E. and A.A.; visualization, D.R. (Daniel Rozban) and A.E.; supervision, A.A., A.E. and G.K.; project administration, A.E., G.K. and A.A.; funding acquisition, A.A., A.E. and G.K. All authors have read and agreed to the published version of the manuscript.

**Funding:** This research received no external funding.

**Data Availability Statement:** Data are contained within the article.

**Conflicts of Interest:** Authors Gil Kedar and Ariel Etinger were employed by the company Ceragon Networks Ltd. The remaining authors declare that the research was conducted in the absence of any commercial or financial relationships that could be construed as a potential conflict of interest.

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
