# Peer review of "Design and Measurement of a Two-Dimensional Beam-Steerable Metasurface for Ka-Band Communication Systems"

_electronics, doi:10.3390/electronics13101998_

Round 1

Reviewer 1 Report

Comments and Suggestions for Authors

1:The reason for the deviation between the scanning angle of the directional pattern test result in Figure 11(a)and the actual simulation is not explained in the text, and the difference between the measured and simulated scanning angles is irregular as the scanning angle increases.

2:Why does the directional pattern in Figure 13 exhibit distortion when scanning at a small angle.

3:A complete description has been provided for the structure adopted in this article, including design considerations, manufacturing processes, and experimental results. The consistency between physical testing and simulation results perfectly validates the design implementation in this article. If the design structure of the text can be compared with a certain number of published literature to highlight the superiority of the work, then the value of the design structure in this article will be further enhanced

Comments on the Quality of English Language

The overall length and grammar of some parts need to be improved, and the layout and aesthetics of the illustrations need to be further refined

Author Response

The authors would like to thank the reviewers for their constructive and detailed remarks. We address and correct accordingly all the reviewer’s remarks. In the revised manuscript, the authors added two tables. One table (Table 2) gives the reader better understanding of the 2D steering mechanism which is the highlight of this study. In addition Figure 6 was improved as well for better understanding the 2D steering. The second table ( Table 3) compares the performance of the proposed reconfigurable MS prototype to other reconfigurable MSs performances in the literature.   Please find the detailed of the corrections to the manuscript according to the reviewers commets. Responses are written in blue. We hope you will find our manuscript suitable for publication in Electronics Journal.

Reviewer 1:

  1. The reason for the deviation between the scanning angle of the directional pattern test result in Figure 11 (a)and the actual simulation is not explained in the text, and the difference between the measured and simulated scanning angles is irregular as the scanning angle increases.

The article has been revised to clarify the disparities observed in the results illustrated in Figure 10(a) and Fig 11 and Fig. 12,   (See  added section below Fig. 12)

  1. Why does the directional pattern in Figure 12 exhibit distortion when scanning at a small angle

The article has been revised to clarify the disparities observed in the results illustrated in Figure 10(a) and Fig 11 and in Fig. 12, . small angles are also discused  (See  added section below Fig. 12)

  1. A complete description has been provided for the structure adopted in this article, including design considerations, manufacturing processes, and experimental results. The consistency between physical testing and simulation results perfectly validates the design implementation in this article. If the design structure of the text can be compared with a certain number of published literature to highlight the superiority of the work, then the value of the design structure in this article will be further enhanced

Table 3 including elaboration regarding the performances was added. (See Table 2 and the added text attached to it)

  1. Comments on the Quality of English Language The overall length and grammar of some parts need to be improved, and the layout and aesthetics of the illustrations need to be further refined

We revisited the article and implemented corrections to enhance the quality of English writing.  {See the entire revised manuscript)

Reviewer 2 Report

Comments and Suggestions for Authors

In this manuscript, the beam scanning metasurface sub reflector is proposed and designed. In my opinion, the manuscript is well organized with attractive performance. In the following, I have several comments or suggestions.

1) The performance of the metasurface sub should be given, such as the input impedance, radiation pattern, efficiency, gain, and so on, which are important for the realization of advanced performance for the reflector.

2) How about the spacing between these elements along two directions, which is important for the sidelobe reduction and mutual coupling.

3) What is the meaning of RCS in these radiation pattern, is it the gain or RCS for the antenna?

4) How about the aperture efficiency of the antenna, which should be given in the manuscript.

5) The authors should give a comparison table to illustrate the advantage and innovation of this designed antenna.

6) The English writing should be carefully improved.

Comments on the Quality of English Language

In this manuscript, the beam scanning metasurface sub reflector is proposed and designed. In my opinion, the manuscript is well organized with attractive performance. In the following, I have several comments or suggestions.

1) The performance of the metasurface sub should be given, such as the input impedance, radiation pattern, efficiency, gain, and so on, which are important for the realization of advanced performance for the reflector.

2) How about the spacing between these elements along two directions, which is important for the sidelobe reduction and mutual coupling.

3) What is the meaning of RCS in these radiation pattern, is it the gain or RCS for the antenna?

4) How about the aperture efficiency of the antenna, which should be given in the manuscript.

5) The authors should give a comparison table to illustrate the advantage and innovation of this designed antenna.

6) The English writing should be carefully improved.

Author Response

The authors would like to thank the reviewers for their constructive and detailed remarks. We address and correct accordingly all the reviewer’s remarks. In the revised manuscript, the authors added two tables. One table (Table 2) gives the reader better understanding of the 2D steering mechanism which is the highlight of this study. In addition Figure 6 was improved as well for better understanding the 2D steering. The second table ( Table 3) compares the performance of the proposed reconfigurable MS prototype to other reconfigurable MSs performances in the literature.   Please find the detailed of the corrections to the manuscript according to the reviewers commets. Responses are written in blue. We hope you will find our manuscript suitable for publication in Electronics Journal.

In this manuscript, the beam scanning metasurface sub-reflector is proposed and designed. In my opinion, the manuscript is well organized with attractive performance. In the following, I have several comments or suggestions

  1. The performance of the metasurface sub should be given, such as the input impedance, radiation pattern, efficiency, gain, and so on, which are important for the realization of advanced performance for the reflector.

Acknowledging the potential for confusion stemming from the article title, abstract, and the initial placement of Figure 1, we understand that readers may have inferred a focus on the design of a sub-reflector within a Cassegrain antenna projection system rather than MS prototype. However, our research diverges from this assumption, centering instead on the design and enhancement of a metasurface reflector optimized for Ka-band frequencies. Accordingly, we have modified the title and abstract and moved Figure 1 to the conclusion section. Here, it clarifies the potential application of the metasurface within sub-reflector contexts, providing insight into future directions for its utilization.

  1. How about the spacing between these elements along two directions, which is important for the sidelobe reduction and mutual coupling.

In this metasurface design, the spacing between unit cells is governed by the dimensions specified in Fig 1, Fig. 2, and Table 1, which describe the features of the designed unit cell. The optimization of this unit cell design focuses on achieving maximum phase deviation and minimal attenuation while considering the overall capacitance values of the VARICAP diode.

  1. What is the meaning of RCS in these radiation pattern, is it the gain or RCS for the antenna?

The measurements in this paper were conducted utilizing a 2-port network analyzer with antennas on each port, enabling analysis of transmission between port 1 and port 2. In CST simulations, the term Radar Cross Section (RCS) is employed instead of the reflection coefficient (RC). The widely accepted term in the majority of published studies concerning metasurfaces is "normalized radar cross section." Therefore, in this article, we adopted this terminology. References to relevant literature employing this terminology can be found in [].

  1. How about the aperture efficiency of the antenna, which should be given in the manuscript.

On page 5, line 132, the type of antennas used in this study is indicated. Specifically, broadband horn antennas of the double-ridge type, model PowerLOG 40400 manufactured by the Aaronia company, were employed. These antennas served for both transmission and reception of signals from the metasurface.

  1. The authors should give a comparison table to illustrate the advantage and innovation of this designed antenna.

We have incorporated Table 3, delineating the advantages of the research presented in this paper in comparison to other studies conducted in the Ka-band frequency range. It is noteworthy that there exists limited literature on metasurfaces operating within the Ka-band frequency range while offering continuous adjustment of the steered beam angle. Furthermore, we have appended a section immediately following Table 3 to elaborate on these benefits.

  1. The English writing should be carefully improved.

We revisited the article and implemented corrections to enhance the quality of English writing.  {See the entire revised manuscript)

Reviewer 3 Report

Comments and Suggestions for Authors

This article elaborates on the  “ 2-D steerable metasurface sub-reflector for the Ka-band  Cassegrain antenna”. As noted by the Authors the article elaborates on the (page 2): “..tuning element losses, substrate losses, DC control, and the isolation between the RF signal and DC”. My comments are as follows:

1) The article main weakness is the missing justification of its novel contribution. Clarify what is is exactly new.

2) The design of the sub-reflector is expected to provide a beam steering of the main reflector beam (Fig.1). However, the study (Fig.6), the measurements setup (Fig.9) and the prototype measurements (Fig.11 to 13) elaborate only on steering the beam of the sub-reflector. The question is whether the produced sub-reflector beams provide the appropriate illumination of the main reflector and in turn the steering of its main beam. Additionally, one should ensure that these sub-reflector  beams do not produce unacceptable spillover effect.

3) A very curious phenomenon is depicted in Figs. 11, 12. Namely, the broadside beam has almost 5 dB higher gain as expected but all other beams have the same gain. On the contrary the gain is expected to decrease away from broadside, as for example in Fig.13.

4) The measurement set-up of Fig.9 should be better explained, as to clarify (in the text) how the beam is steered in two orthogonal directions.

Comments on the Quality of English Language

5) The manuscript needs a grammatical and syntactical revision.

Author Response

The authors would like to thank the reviewers for their constructive and detailed remarks. We address and correct accordingly all the reviewer’s remarks. In the revised manuscript, the authors added two tables. One table (Table 2) gives the reader better understanding of the 2D steering mechanism which is the highlight of this study. In addition Figure 6 was improved as well for better understanding the 2D steering. The second table ( Table 3) compares the performance of the proposed reconfigurable MS prototype to other reconfigurable MSs performances in the literature.   Please find the detailed of the corrections to the manuscript according to the reviewers commets. Responses are written in blue. We hope you will find our manuscript suitable for publication in Electronics Journal.

Reviewer 3:

This article elaborates on the “ 2-D steerable metasurface sub-reflector for the Ka-band Cassegrain antenna”. As noted by the Authors the article elaborates on the (page 2): “..tuning element losses, substrate losses, DC control, and the isolation between the RF signal and DC”.

  1. The article's main weakness is the missing justification of its novel contribution. Clarify what is exactly new.

We have integrated Table 2, outlining the advantages of the research presented in this article compared to other studies conducted in the Ka-band frequency range. The novelties presented in this paper are as follows:

  1. a) Limited existing literature on metasurfaces operating in the Ka-band frequency range while allowing continuous adjustment of the steered beam angle. Consequently, this paper is unique and could significantly contribute to this frequency band. (See Table 2 and the attached text following it)
  2. b) Only a few studies demonstrate beam steering in both 1-D and 2-D with a fully automatic measurement setup that accurately measures the steering angles. Figure 6 demonstrates and explaines this unique mechanism measurement setup.
  3. c) Presentation of measurement results validating the ability to continuously steer a beam in the Ka-band using varactors. (see elaboration below Figure 6)

  1. The design of the sub-reflector is expected to provide a beam steering of the main reflector beam (Fig.15). However, the study (Fig.6), the measurements setup (Fig.9), and the prototype measurements (Fig.11 to 13) elaborate only on steering the beam of the sub-reflector. The question is whether the produced sub-reflector beams provide the appropriate illumination of the main reflector and in turn the steering of its main beam. Additionally, one should ensure that these sub-reflector beams do not produce an unacceptable spillover effect.

Acknowledging the potential for confusion stemming from the article title, abstract, and the initial placement of Figure 1, we understand that readers may have inferred a focus on the design of a sub-reflector within a Cassegrain antenna projection system. However, our research diverges from this assumption, centering instead on the design and enhancement of a metasurface reflector optimized for Ka-band frequencies. Accordingly, we have rectified the title and abstract and relocated Figure 1 to the conclusion section. Here, it clarifies the future potential application of the metasurface within sub-reflector contexts, providing insight into future directions for its utilization.

A very curious phenomenon is depicted in Figs. 10 11. Namely, the broadside beam has almost 5 dB higher gain as expected but all other beams have the same gain. On the contrary the gain is expected to decrease away from broadside, as for example in Fig.12.

Utilizing varactors in a metasurface introduces losses primarily originating from its resistive component. When steering the beam, a greater number of unit cells experience attenuation, particularly around the resonance frequency of the unit cell. This results in an overall increase in the total attenuation of the reflected beam, as depicted in Figures 11, 12, and 13. The CST simulations depicted in these figures were further validated by measurements (See the text below Fig. 10  and especially Fig. 12) . Other studies have shown similar results [33,34]

  1. The measurement set-up of Fig.8 should be better explained, as to clarify (in the text) how the beam is steered in two orthogonal directions.

Figure 6 was improved to better understanding of the method of steering the beam in 1D and 2D. Additionally, we have incorporated Table 2, which describes how the different phases in the metasurface array were selected to achieve 2D beam steering. Furthermore, an explanation has been added below Figure 6, detailing how beam steering is achieved in both 1D and 2D configurations.

 Comments on the Quality of English Language. The manuscript needs a grammatical and syntactical revision.

We endeavored to rectify the grammatical and syntactical errors in the article, The layout will be fixed in the final version.(See the entire revised manuscript)

Round 2

Reviewer 3 Report

Comments and Suggestions for Authors Dear Authors, Thank you for carefully addressing all of my comments and concerns.